# COMMIT: Consideration of metabolite leakage and community composition improves microbial community reconstructions

**Philipp Wendering**[1], **Zoran Nikoloski**[1,2]*

**1** Bioinformatics, Institute of Biochemistry and Biology, University of Potsdam, Potsdam, Germany,
**2** Systems Biology and Mathematical Modeling, Max Planck Institute of Molecular Plant Physiology, Potsdam, Germany

* nikoloski@mpimp-golm.mpg.de

**Data Availability Statement:** The data and source code generated for this manuscript are available at https://doi.org/10.5281/zenodo.6334079. Draft and consensus reconstructions from the individual

## Abstract

Composition and functions of microbial communities affect important traits in diverse hosts, from crops to humans. Yet, mechanistic understanding of how metabolism of individual microbes is affected by the community composition and metabolite leakage is lacking. Here, we first show that the consensus of automatically generated metabolic reconstructions improves the quality of the draft reconstructions, measured by comparison to reference models. We then devise an approach for gap filling, termed COMMIT, that considers metabolites for secretion based on their permeability and the composition of the community. By applying COMMIT with two soil communities from the *Arabidopsis thaliana* culture collection, we could significantly reduce the gap-filling solution in comparison to filling gaps in individual reconstructions without affecting the genomic support. Inspection of the metabolic interactions in the soil communities allows us to identify microbes with community roles of helpers and beneficiaries. Therefore, COMMIT offers a versatile fully automated solution for large-scale modelling of microbial communities for diverse biotechnological applications.

## Author summary

Microbial communities are important in ecology, human health, and crop productivity. However, detailed information on the interactions within natural microbial communities is hampered by the community size, lack of detailed information on the biochemistry of single organisms, and the complexity of interactions between community members. Metabolic models are comprised of biochemical reaction networks based on the genome annotation, and can provide mechanistic insights into community functions. Previous analyses of microbial community models have been performed with high-quality reference models or models generated using a single reconstruction pipeline. However, these models do not contain information on the composition of the community that determines the metabolites exchanged between the community members. In addition, the quality of metabolic models is affected by the reconstruction approach used, with direct consequences on the inferred interactions between community members. Here, we use fully

approaches are available at https://doi.org/10.5281/zenodo.6334097.

**Funding:** The authors received no specific funding for this work.

**Competing interests:** The authors have declared that no competing interests exist.

automated consensus reconstructions from four approaches to arrive at functional models with improved genomic support while considering the community composition. We applied our pipeline to two soil communities from the *Arabidopsis thaliana* culture collection, providing only genome sequences. Finally, we show that the obtained models have 90% genomic support and demonstrate that the derived interactions are corroborated by independent computational predictions.

## Introduction

Microbial communities have been extensively studied due to their importance in ecology [1,2], human health [3,4], and biotechnological applications [5]. It has also been suggested that microorganisms have particular roles in a community; for instance, the Black Queen (BQ) hypothesis [6] suggests the existence of BQ functions, such as production of membrane-permeable products, that are essential for members termed helpers, but unavoidably available to other community members, termed beneficiaries. These roles are achieved by active transport and/or leakage of diverse metabolites, including biomass precursors [7].

Constraint-based modelling of genome-scale metabolic networks provides the means to in silico analyse microbial community interactions [8–12]. The existing metabolic reconstruction approaches [13–19] rely on linking genome annotation to enzymatic reactions from various databases [19–23]. A structural comparison of the metabolic models resulting from these approaches showed that the portions of shared reaction, metabolite, and gene sets are rather moderate [24]. Hence, a consensus reconstruction could provide a means to combine the advantages of the existing approaches [25,26].

However, a consensus reconstruction is not guaranteed to be functional (i.e. to simulate growth), as knowledge gaps may occur in all of the underlying (draft) reconstructions. To address this issue, several algorithms for gap filling, applicable to single and community reconstructions [13], have been proposed; however, they are not feasible to gap fill reconstructions of large communities (as described e.g. in [27,28]) due to the sheer amount of computational resources required. Since members of microbial communities are often dependent on each other, gap-filling solutions of the community members must also be put into context of the community in which the organisms co-exist. In addition, the usage of gap-filled solutions and exudates from other community members may further reduce the overall number of added reactions to fill in the gaps in the metabolic reconstruction for the community. Existing computational efforts have shown that the gap-filling medium as well as the order in which gap-filling is applied plays a very important role in the reconstructed model [29].

The problem of finding metabolic interactions within microbial communities using constraint-based methods has been addressed by multiple studies, focussing on pairwise or higher-level interactions [8,10,30–34]. Although pairwise interactions within large communities can be well described both qualitatively and quantitatively [10,31], the results may, however, be inaccurate when large microbial communities are considered where interactions become more complex. Approaches like SteadyCom [34] or MICOM [35] overcome this by performing whole-community optimization at the cost of high computational effort due to the size of the arising constraint-based problems. In SteadyCom, the number of iterations needed to approximate the community growth rate with a sufficient accuracy is independent of the number of organisms. However, the linear problem that is solved during each iteration increases with the number of organisms, making it challenging to apply this approach to large communities. In contrast, the MICOM approach has been successfully applied to large communities [35]. Importantly, these fully automated approaches rely on: (1) high-quality models

available for all community members, and (2) transport reactions already contained in the model, and thus neglect permeability to define the set of metabolites that can be exchanged between community members.

Here, we describe COMMIT, a constraint-based approach that respects the composition of the microbial community and metabolite leakage in the process of gap filling metabolic reconstructions of the respective community members. We then apply COMMIT to high-quality consensus metabolic reconstructions based on genomes of the isolates from the *Arabidopsis thaliana* microbial culture collections, At-SPHERE, for two different soil community compositions [36,37]. Altogether, our results demonstrate that the consensus approach in combination with the gap filling approach that respects the community composition renders COMMIT a valuable addition to the approaches for fully automated reconstruction of large-scale metabolic reconstructions of microbial communities.

## Results

### Draft genome-scale metabolic reconstructions for 432 isolates in At-SPHERE show substantial structural differences

We used the high-quality draft genomes for 432 isolates available from At-SPHERE [38] to obtain their draft genome-scale metabolic reconstructions. To this end, we applied four widely-used, fully automated approaches for metabolic network reconstruction relying on few parameters, namely: KBase [19], CarveMe [14], RAVEN 2.0 [17], and AuReMe/ Pathway Tools [15,16] (for reviews, see [24,39]). We then converted the draft reconstructions generated by the four approaches to a common format by multiple technical adaptations (see Materials and Methods).

To compare the structure of the reconstructions after the conversion, we employed eight distance measures, including: the Jaccard distance based on the sets of metabolites, reactions, E.C. numbers, genes, and dead-end metabolites, the number of dead-end metabolites, the rank correlation of E.C. number occurrence, usage of cofactors, and the SVD distance of the stoichiometric matrices (see S1 Text). We then combined the resulting distance matrices in a compromise distance matrix for each isolate; to facilitate a global comparison of the different approaches, we calculated the compromise distance matrix per approach based on the isolate-specific compromise matrices determined before (see Materials and Methods, Fig 1A). The structural comparison revealed substantial differences across the draft genome-scale metabolic reconstructions generated by the four approaches (Fig 1A). In the compromise distance matrix obtained from the eight distance measures across all isolates, the draft reconstructions showed an average distance of 0.64 to each other, ranging from 0.54 to 0.72 (with 1 denoting the largest difference). Regarding the gene identifiers, the reconstructions generated using KBase, CarveMe, and RAVEN 2.0 were more similar to each other in comparison to the reconstructions resulting from AuReMe/Pathway Tools.

To determine whether or not the utilized distance measures are biologically relevant, we next calculated the correlation between the distance matrices and sequence distance of the 16S rRNA sequences. We found that the Jaccard distances corresponded to the generated phylogeny, with significant correlations ranging from 0.63 to 0.75 with an average of 0.70 ($p < 0.001$) over the 432 isolates (Fig 1B). We also observed moderate but significant correlations of the SVD distance and the rank correlation of cofactor usage with sequence distance ($\bar{\rho} = 0.25, \; p < 0.001$). In contrast, the occurrence of dead-end metabolites showed a very low correlation with sequence distance ($\bar{\rho} = 0.04$). These findings indicated that the utilized distance measures to quantify the structural differences between the draft reconstructions are biological relevant, further supporting our result that the reconstructions obtained by some of the utilized approaches are strikingly different (Fig 1A).

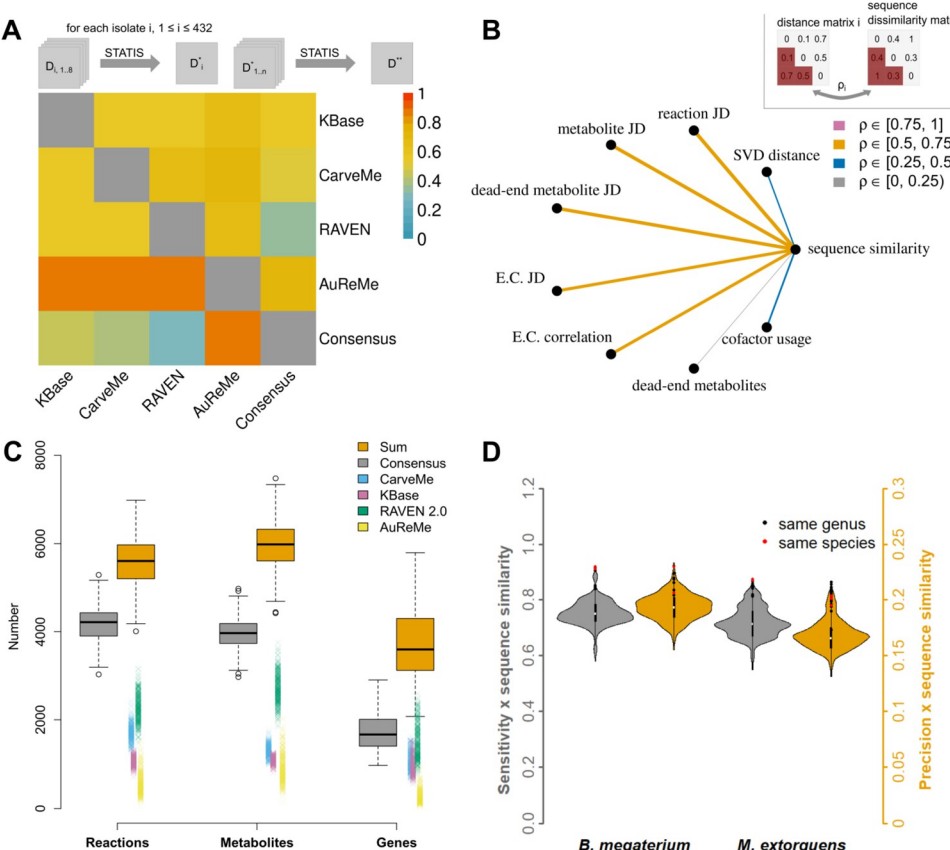

**Fig 1. Structural comparisons of draft and consensus reconstructions. (A)** Structural differences between the reconstructed draft reconstructions and their consensus. upper triangle: For each isolate i (1≤i≤432) eight distance matrices ($D_j^i$, 1≤j≤8, see Materials and Methods) were computed for reconstructions from KBase [13], CarveMe [14], RAVEN 2.0 [17], AuReMe/ Pathway Tools [15,16], and their consensus. These matrices were then combined for isolate i by finding a compromise matrix $D_i^*$ (STATIS method [40–42]). The resulting compromise matrices were again combined using STATIS, resulting in a distance matrix $D^{**}$ capturing the overall difference between pairs of approaches. lower triangle: Gene Jaccard distances between reconstructions across all four methods and the consensus, with compromise distance matrices calculated as for the upper triangle. **(B)** The same eight distance measures from (A) were used for pairwise comparisons of consensus reconstructions between all isolates (n = 432). Each of the resulting distance matrices was compared to sequence distance of the 16S rRNA sequences using Mantel correlation (ρ). The widths and colour coding of the lines connecting the distance measures with sequence similarity indicate the strength of the correlation. **(C)** Comparison of consensus reconstructions to draft reconstructions. The numbers of reactions, metabolites, and genes are shown for the consensus in comparison to the single draft reconstructions generated by the four approaches. To depict the overlap between the four approaches, the sum of the numbers of reactions, metabolites, and genes per reconstruction is shown ("Sum", orange box) as the hypothetical number of these features if no reconciliation has happened. **(D)** Similarity of consensus reconstructions to selected reference models. The sensitivity (grey, left) and precision (orange, right) with respect to metabolite and E.C. number sets were calculated for each of the 432 reconstructions and each reference model. These values were scaled by the sequence similarity to the 16S rRNA sequences of the used references. Isolates that were assigned the same genus (9 for *B. Megaterium* and 27 for *M. extorquens*) or species (2 for *B. Megaterium* and 5 for *M. extorquens*) according to Bai et al. [38], are shown as black or red dots.

## Consensus metabolic reconstructions show high organism specificity

It has been demonstrated that the integration of multiple metabolic reconstructions into a consensus reconstruction leads to a reduced number of blocked reactions due to the complementarity of their information content [25,43,44]. Since we observed that the draft reconstructions generated with the four approaches differed in their underlying genome annotation and

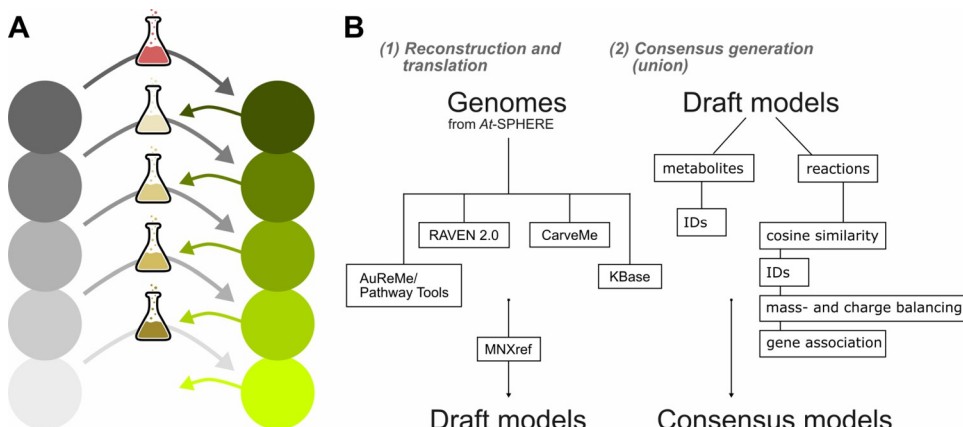

**Fig 2. Schematic workflow of metabolic model reconstruction from multiple approaches. (A)** The consensus models were gap-filled conditional on the community composition according to COMMIT. Grey circles represent the reconstructions for every organism before gap filling. Green color indicates functional models after gap filling. The medium for the first reconstruction was the respective auxotrophic medium predicted using KBase [13] (red medium). For subsequent reconstructions, a minimal medium was used, which was enriched by the secreted metabolites of already gap-filled models (green arrows). **(B)** Annotated genomes from At-SPHERE were used as the basis to reconstruct 432 draft metabolic reconstructions using the four recent methods from KBase [13], CarveMe [14], RAVEN 2.0 [17], and AuReMe/Pathway Tools [15,16]. These draft reconstructions were merged into consensus reconstructions per isolate.

downstream reaction and metabolite sets (Fig 1A), we hypothesized that there would be an overall increase in quality as well as a decrease in the number of gaps (on the path to biomass production) in a corresponding consensus reconstruction. The consensus generation consisted of matching metabolite, reaction, and gene identifiers (Fig 2B). Since the metabolites in the MetaNetX database are already structurally matched between various databases, duplicate metabolites could be removed from the consensus by only considering their identifiers. We employed cosine similarity to identify reactions of similar stoichiometry that may have opposite directions, lack protons or whose coefficients differ by a factor. Further, we compared mass-balance, reversibility, direction, and protonation. Previously published approaches as COMMGEN [44] or MetaMerge [25] were not applicable since they do not support the current MNXref format used in the MetaNetX database or are no longer maintained.

We found that the consensus reconstruction is considerably smaller and, thus, does not correspond to the sum of the number of reactions, metabolites, and genes contained in the underlying draft reconstructions (Fig 1C). Further, the proportions of reactions, metabolites, and genes were not uniform across the draft reconstructions obtained from the different reconstruction approaches. The RAVEN 2.0 reconstructions consistently included larger number of reactions, metabolites, and genes compared to all other methods, with AuReMe\Pathway Tools draft reconstructions exhibiting the lowest sums of these properties (Fig 1C). We found that an average of 41.5% of reaction in RAVEN 2.0 reactions was duplicated after translation to MNXref namespace. Hence, incomplete matching between KEGG and MetaCyc namespaces could provide an explanation for the increased size of these models. The draft reconstructions from RAVEN 2.0 exhibited the smallest overall distance (0.37) to the consensus reconstructions, while the draft reconstructions obtained from the other tools showed an average distance of 0.59 to the consensus (Fig 1A). The Jaccard distances of gene identifiers to the consensus reconstruction ranged between 0.31 for RAVEN 2.0 reconstructions and 0.87 for AuReMe reconstructions. Despite the observed differences in reaction, metabolite, and gene sets, the overall distribution of metabolic subsystems (assessed by KEGG ko09100 top-level definitions) was similar across all draft models and for the resulting consensus models.

To show the organism-specificity of the resulting consensus reconstruction, next we assessed the similarity of the consensus to selected reference models for isolates that were resolved down to species classification, namely *Bacillus megaterium* (iMZ1055) and *Methylobacterium extorquens* (iRP911). To this end, we employed the manually-curated models for these two species [45,46] and first translated their identifiers into the MNXref namespace (only possible for metabolites), facilitating comparison of metabolites and E.C. numbers. Every consensus reconstruction was compared to each of the two curated models to determine the number of true positives, false positives, and false negatives and to calculate sensitivity and precision (see Materials and Methods). Both values were scaled by the sequence similarity of an isolates' 16S rRNA sequence to the sequence of the reference species (S1 Fig). We found that the consensus reconstructions for isolates of the same species were more similar to the curated models than reconstructions for isolates that share only the genus or reconstructions of even less-related isolates (Fig 1D). Notably, most of the reconstructions for isolates with only matching genus exhibited higher precision for *M. extorquens* than the ones from the same species. Moreover, the consensus reconstructions showed higher values for scaled sensitivity compared to the individual draft reconstructions (S2 Fig). In contrast, the values for the scaled precision of the consensus reconstructions were smaller than those of the draft reconstructions. This can be explained by the overlap of the draft reconstructions with respect to the sets of features (i.e. metabolites and E.C. numbers): The number of the non-overlapping features (i.e. false positives) increases more than the number of features matching with the reference models (i.e. true positives) during the consensus generation. Hence, the precision is lower for the consensus as the ratio of true positives to false positives is shifted. Yet, across all draft reconstructions, except for AuReMe\Pathway Tools, reconstructions for the same species and genus were more similar to each other than reconstructions of phylogenetically more distant isolates. Therefore, we concluded that the consensus of metabolic reconstructions shows a higher organism specificity as well as a higher quality than the draft reconstructions obtained from the individual approaches.

## COMMIT provides gap-filling that respect the community composition and metabolite leakage

Constraint-based analysis of microbial communities is based on assembling functional metabolic models for members of the community that can, in turn, be used to simulate microbial growth. However, obtaining a functional metabolic model from a draft reconstruction entails adding reactions to an incomplete metabolic network to enable simulation of growth [47,48]; often the inserted reactions are without genomic support.

The existing gap-filling methods are applied with reconstructions of individual species and do not consider the innate dependence between the community members [6,9,49]. In contrast, our approach, termed COMMIT, aims to identify a minimal gap-filling solution that respects the composition of the microbial community. To this end, COMMIT explores a specified number of random orderings of the community members. Draft reconstructions for each member of the community are then gap-filled following a given random ordering of the community members. COMMIT relies on the FastGapFilling algorithm, which optimizes a weighted sum of the fluxes through the biomass and additional candidate reactions by using linear programming (LP) formulation (see Materials and Methods section for more detail).

In the following, we will use the terms permeable metabolites for metabolites that are allowed to be exported from the model with respect to their membrane permeability and minimal growth reduction. Further, secreted metabolites are those metabolites that are actually exported from the model and exchanged metabolites are not only secreted by one or multiple models but also taken up by other models.

COMMIT starts with a minimal medium; secreted metabolites from gap-filled (i.e. functional) models are then determined and used to enlarge the medium, thereby shaping the overall gap-filling solution over the course of each random ordering. The orderings are compared with respect to four criteria, including: (i) the total number of added reactions, (ii) dependence of the first member on secreted metabolites of subsequent models, (iii) the overall number of permeable metabolites, and (iv) the sum of biomass fluxes of all community members (see Materials and Methods). We determined permeable metabolites after gap-filling each of the draft reconstructions following the given random ordering of community members. A metabolite is allowed to be secreted (and is considered permeable in the respective reconstruction), if its export does not decrease the flux through the biomass reaction by more than a pre-specified threshold. We rely on using a threshold value (of 10%) because an overly altruistic behaviour of single community members is not considered biologically relevant. However, when no decrease in biomass production is allowed, the number of permeable metabolites only changed by 3.6%, likely because sink reactions for intracellular metabolites can also increase the biomass production.

The set of permeable metabolites is in turn made available to the subsequent reconstructions by allowing a lower cost for their import, thus enlarging the initial gap-filling medium (Fig 2A). Assuming that the community members depend on each other [6], we hypothesized that the enlarged medium is expected to reduce the added reaction sets of subsequent members of the community. Further, the existence of an optimal ordering is likely, since gap-filling solutions do not only depend on the algorithm and candidate reactions employed, but also on the underlying medium [29]. Given an ordering of the isolate reconstructions, the solution for the first reconstruction in the ordering is obtained by using its auxotrophic medium as determined using KBase [13]. This strategy was employed to prevent an unrealistically high number of added reactions for the first reconstruction assuming the included compounds, most importantly essential amino acids, are likely produced by other community members. To this end, we respected the dependence of the first model on permeable metabolites from the remaining models in the choice of the optimal ordering. Altogether, the gap-filling procedure is conditional on the community composition, as it takes metabolic capabilities of all community members into consideration.

In addition to the medium, the choice of candidate reactions affects the set of reactions that are added to close gaps in biological pathways [50]. In COMMIT, the objective of the gap-filling LP includes specific costs for different reaction types (e.g. transport reactions for highly-permeable metabolites and reactions with sequence evidence). To predict metabolite permeability, we obtained molecular properties from PubChem [51] (see Materials and Methods section for details). Altogether, six parameters were used for permeability prediction based on Lipinski's rule [52,53], resulting in reduced costs for 2534 out of 4520 transport reactions in the gap-filling database. Moreover, reactions assigned to enzymes with sequence evidence in the respective genome were assigned a lower cost (see Materials and Methods).

To assess the performance of the approach, we applied COMMIT to a well-described two-species community of Desulfovibrio vulgaris and Methanococcus maripaludis [54–56]. We removed reactions randomly from both networks (1%, 2%, 5%, and 10%) to see how many of those COMMIT would recover with considering the community composition and without. We repeated this process 50 times and calculated the precision and recall values (S3 Fig). The results from this analysis indicated that gap-filling in the context of the community, as performed in COMMIT, results in slightly improved values for the precision and recall in comparison to applying the gap-filling of individual reconstructions.

## COMMIT results in high-quality soil microbial communities based on consensus reconstructions from At-SPHERE

We applied COMMIT to two soil microbial communities whose composition was determined by comparison of 16S rRNA sequences from environmental samples with those from the cross-reference operational taxonomic units (OTU) used in At-SPHERE [38]. This resulted in 20 and 24 recovered soil isolates from two respective studies, referred to as "Bulgarelli" [37] and "Schlaeppi" [36]. The analysis revealed the existence of high- and low-abundant isolates in both experiments (Tables A and B in S1 Text). The draft metabolic reconstructions for the members of the two communities were gap-filled by applying COMMIT with 100 iterations, corresponding to different random orderings, and an initial gap-filling medium containing only D-glucose as a carbon source (Table C in S1 Text).

The explored orderings were compared with respect to the number of added reactions, biomass fluxes, and number of permeable metabolites, after scaling by the respective value in the optimal solutions (using the four abovementioned criteria). Since solutions with the smallest number of added reactions across all iterations are generally favoured, all suboptimal solutions have a larger number of added reactions (Fig 3). Moreover, for both communities, we observed a shift in the number of secreted metabolites when comparing the optimal with the suboptimal solutions from the other random orderings. Most of the suboptimal solutions exhibited a smaller number of permeable metabolites for all reconstruction types, except for the KBase reconstructions in the Schlaeppi community (Fig 3). This can be explained by the fact that exchange reactions for secreted metabolites will render the solution set of added reactions smaller as a certain metabolite can be taken up instead of adding reactions for its synthesis. In contrast, such a pronounced trend could not be observed for the sums of predicted growth rates.

We next used the memote test suite to assess the quality of the resulting functional models. The generated models of the Schlaeppi community exhibited an average score of 35% (Table D in S1 Text). A strong reduction in the memote score was expected due to missing metabolite, reaction, and gene annotation by databases other than MetaNetX. Nevertheless, the consensus models show an improvement in total score by 12% and 19% to AuReMe\Pathway Tools and CarveMe reconstructions, while it decreased by 1.3% in comparison to KBase draft reconstruction. Moreover, the generated models showed an average consistency of 44%. Importantly, the average fraction of reactions with associated GPR rules in the consensus was 90%, since reactions with genomic evidence were preferred during both the merging and gap-filling procedures. Therefore, the resulting models showed very similar characteristics to the previously used reference models. In comparison, the reference model for *B. megaterium*, iMZ1055, was scored with 25% and 55% consistency. We could not obtain a memote score for the *M. extorquens* model and RAVEN 2.0 reconstructions, due to technical issues with the software.

## COMMIT shapes and significantly reduces the gap-filling solution

The observed differences between different random orderings of community members indicated that there exists an optimal solution that minimizes the number of added reactions for the particular community composition. Next, we were interested to assess the differences between the gap filling solutions for individual reconstructions with and without respecting the community composition. We found that COMMIT decreased the number of added reactions compared to the individual gap filling of the reconstructions using the same algorithm without allowing for metabolite exchange between the members (Figs 4A and S4A). Despite the low number of added reactions for the consensus reconstructions, we observed, as expected, a significant decrease in the gap-filling solution size with COMMIT in comparison

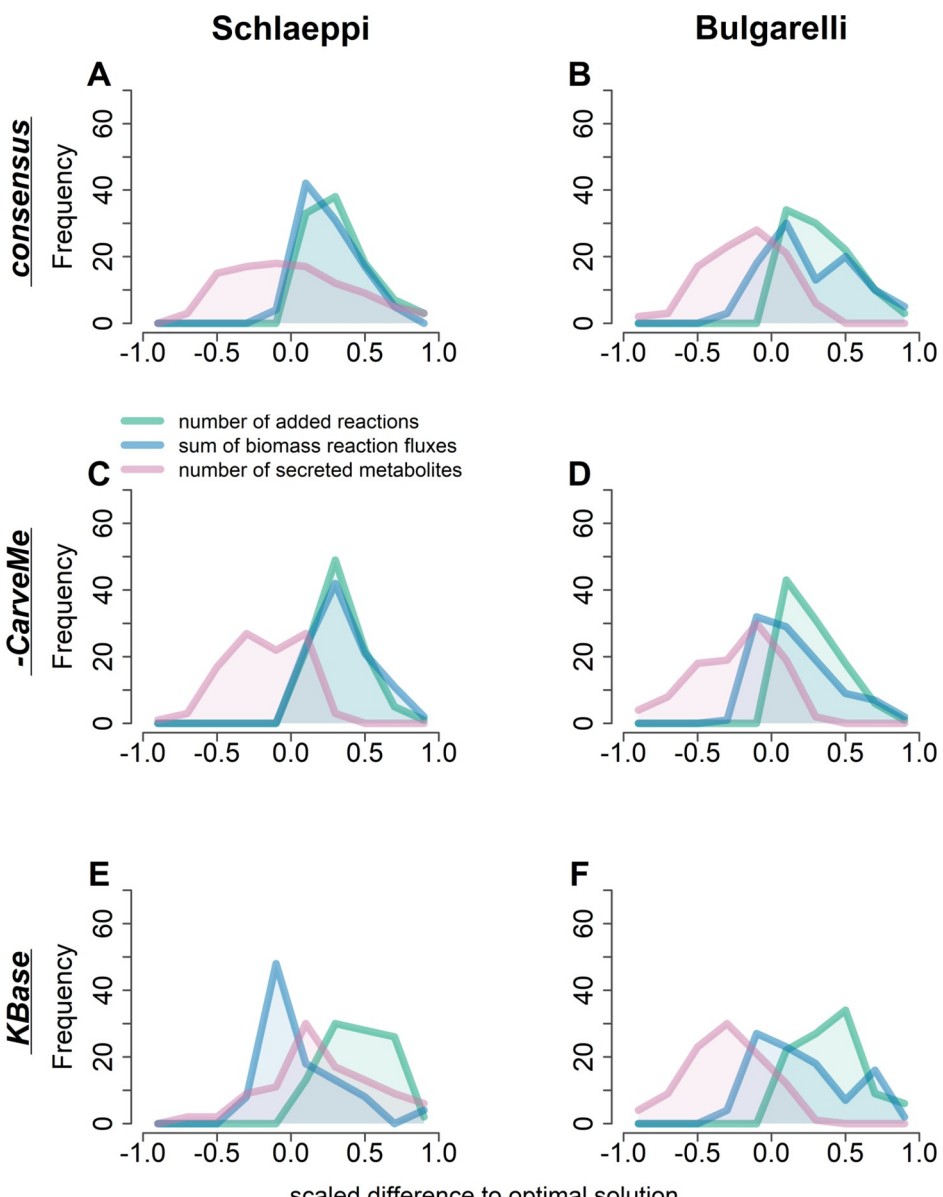

**Fig 3. Differences in gap-filled reconstructions across the random orderings of isolates.** The number of added reactions, sum of biomass fluxes, and the total number of secreted metabolites by the gap-filled reconstructions were compared between all orderings explored by COMMIT. **(A,B)** consensus reconstructions from all approaches **(C,D)** consensus reconstructions without CarveMe draft reconstructions **(E,F)** converted KBase draft reconstructions. Results for the Schlaeppi community are shown on the left and results from the Bulgarelli community on the right panels. The lines represent the number of counts for each histogram bin, for the added reactions, growth rates, and numbers of exchanged metabolites. The values for each measure were scaled to the respective value associated with the optimal ordering. Number of iterations: n = 100; Number of recovered isolates per community: 24 for Schlaeppi and 20 for Bulgarelli.

to the individual gap filling approach (Fig 4A). Notably, the average proportion of added reactions with sequence support did not change between the individual and conditional gap-filling. The proportions of added reactions with sequence support for individual gap-filling were 6.7% for the full consensus, 29.1% for the consensus without CarveMe reconstructions, and 37.1% for KBase draft reconstructions 37.1%.

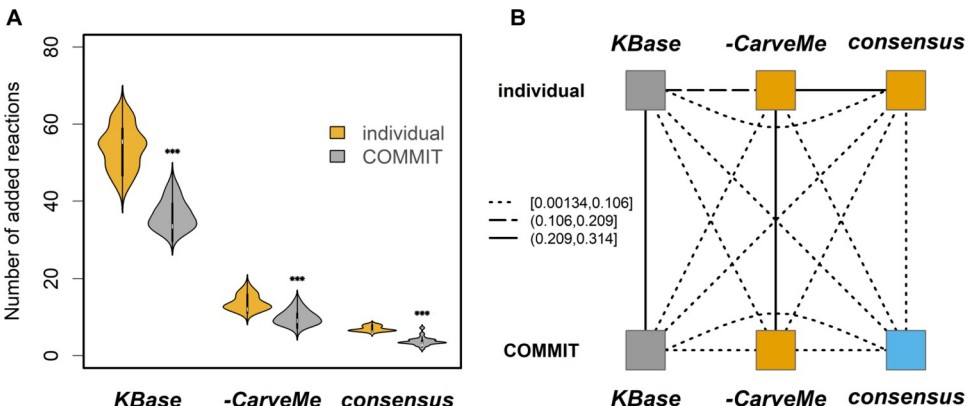

**Fig 4. Comparison of gap filling solutions from COMMIT and individual models for the Schlaeppi community.**
Full consensus (consensus), consensus without CarveMe reconstructions (-CarveMe), and KBase draft reconstructions (KBase) were gap-filled either individually or using the COMMIT approach. **(A)** Sizes of gap-filling solution sets were compared for each reconstruction type using a paired Wilcoxon rank sum test (*** $p < 0.001$). **(B)** Pairwise comparison of added reactions obtained for each reconstruction and gap-filling type by calculating the Jaccard similarity per isolate. The resulting matrices were merged per group using the STATIS method [40–42]. The obtained values were grouped using K-means clustering (K = 3). The line types indicate the average similarity between the compared groups.

To show that the set of added reactions is not only smaller but also differs in its composition, we performed K-means (K = 3) clustering with the matrix of Jaccard similarities of the added reaction sets for the full consensus, consensus without CarveMe, and KBase draft reconstructions gap-filled either with or without considering the community composition (Figs 4B and S4B). We used three clusters based on the hypothesis that the solutions would group together by the reconstruction type (i.e. consensus, consensus without CarveMe, KBase draft reconstructions) rather than by the applied approach (i.e. with and without considering the community composition). We found that the reaction sets added to the KBase draft reconstructions clustered together, while the added sets for the two other reconstruction types formed clusters by method (i.e. COMMIT and individual) rather than by type. Nevertheless, one can observe that that the reaction set that was added to the full consensus reconstruction shows relatively low similarity to all other solutions, most importantly to the individual solution for the same reconstruction type (Fig 4B).

## Secreted metabolites reveal putative metabolic interactions

As the reduction of added reaction sets is based on the metabolic interactions between the models, the underlying set of exchanged metabolites was investigated to find putative dependencies within the investigated communities. To this end, we determined the contribution of each community member to the common pool of exchanged metabolites, without considering those that were already contained in the minimal medium.

First, we grouped the models based on bacterial families as found in At-SPHERE [38]. We observed that most of the models across all families show putative dependencies for $Fe^{2+}$, $Fe^{3+}$, and $SO4^{2-}$, which are in turn also provided by members of most bacterial families (Fig 5, Bulgarelli: S5 Fig). In addition, amino acids, like: L-leucine, L-asparagine, L-isoleucine, L-arginine, and L-lysine were found to be exchanged in the Schlaeppi community but only L-leucine in the Bulgarelli community. Further, we observed that members of the Micrococcaceae appear to have a rather broad spectrum of utilized nutrients, which spans most of the exchanged amino acids and carbohydrates. In contrast, bacterial families, such as:

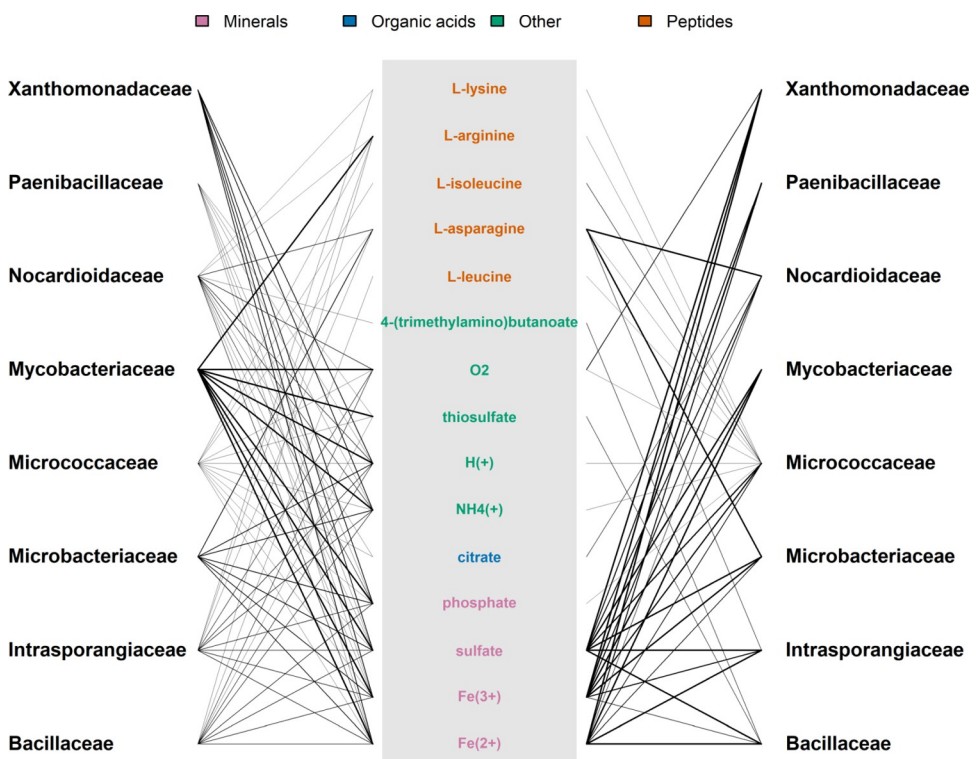

**Fig 5. Putative metabolic interactions between the bacterial families in the Schlaeppi community.** The sets of imported and secreted metabolites that were determined for each member during conditional gap-filling and grouped into corresponding bacterial families. On the left-hand side, metabolite export is shown, which does not require a reduction of growth greater than 10%. The common pool of exchanged metabolites is shown in the center, which were classified using the KEGG BRITE br08001 with manual refinement. On the right-hand side, import reactions are shown, which were introduced during the gap-filling procedure. Line widths represent the number of community members per import/export of a metabolite, scaled by the abundance of the respective family.

Mycobacteriaceae and Paenibacillaceae, were found to depend on a smaller, more specific set of imported metabolites. Although members of most of the bacterial families can export a variety of metabolites at a relatively low cost, there exist families that include more members exporting these metabolites (e.g. Mycobacteriaceae) than others (e.g. Micrococcaceae). Interestingly, L-isoleucine, L-leucine, 4-(trimethylamine)butanoate, L-lysine, L-asparagine, thiosulfate, and citrate were reported to be exchanged between only up to three or within the same families, respectively, indicating specialized interactions. The import of $H^+$ was observed for only 1 community isolate. This finding indicates that the members of a bacterial community prefer different pH in their local environment. This also suggests that almost all members have the ability to increase the acidity of the soil without a large growth compromise.

To analyse directed interactions between the community members in more detail, we conducted a downstream analysis of the gap-filled models using the SMETANA tool [10] (implementation from github.com/cdanielmachado/smetana). For the Schlaeppi community, we obtained a high metabolic interaction potential (MIP) of 0.98, which indicates that the community members can provide most of the essential nutritional components for itself by exchanging metabolites. SMETANA predicted metabolic interactions between 17 out of the 24 community members. To compare the obtained dependencies with our results, we transformed the directed interactions predicted by SMETANA to undirected interactions defined by the pairwise overlap of imported and secreted metabolites. We found that about 43% of all

possible interactions were predicted by both methods. Additionally, 17% of all isolate pairs were predicted to be non-interacting by both methods, summing up to 60% agreement between SMETANA and COMMIT (S6 Fig). However, the intersection of the sets of exchanged metabolites from both methods consisted of only eight metabolites compared to a total of 15 metabolites from COMMIT and 29 from SMETANA; these included, aside from minerals, L-arginine, L-asparagine, L-lysine, and $NH4^+$. Interestingly, SMETANA predicted a much larger set of exchanged metabolites, which further includes more amino acids, organic acids, as well as nucleotides.

To investigate the functional effects of the dependence on the import of metabolites, we investigated whether or not there is a decrease in growth upon blocking the respective uptake reactions for each metabolite. To this end, we calculated the ratio of flux through the biomass reaction to the optimal specific growth rate without any further constraints. Therefore, we performed FBA with additional loop-law constraints and one-norm minimization of fluxes to avoid flux through futile cycles [57]. In addition, we grouped the models based on these ratios using hierarchical clustering (Fig 6). The heatmap shows that only the removal of a few secreted metabolites results in a reduction of growth for the community members that take up the respective compounds. Further, we observed that these dependences do not cluster together by taxonomic class or family. Nevertheless, we observed that many isolates strongly depend on $Fe^{2+}$, $Fe^{3+}$, and sulfate. Several members showed dependences on L-asparagine, L-isoleucine, or citrate. Therefore, our findings show that the fully automated metabolic reconstructions while considering the composition of microbial community and metabolite leakage can be used to identify exchanged metabolites and metabolic dependences of the individual isolates.

## Discussion

Functional separation of the metabolic capabilities in a microbial community represents an important factor directing ecosystem functioning. While ecological interactions describe the effects that species can have on one another they miss out on complex metabolic relationships

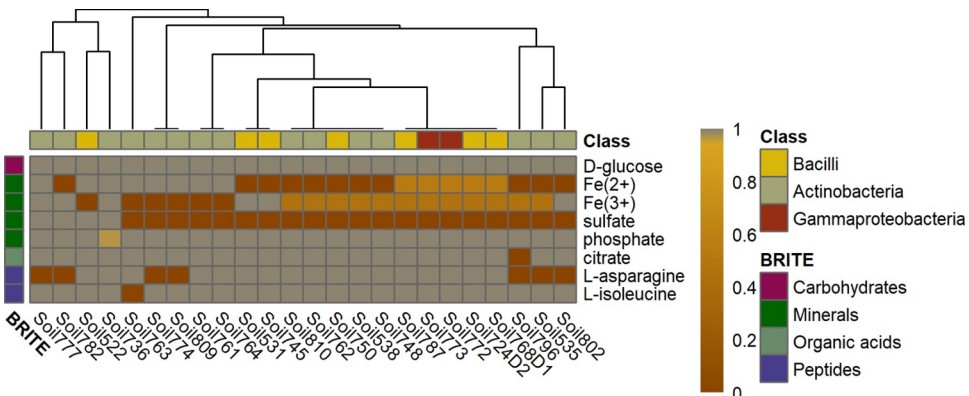

**Fig 6. Reduction of biomass flux upon blocking the uptake of metabolites whose import was added during conditional gap filling of the Schlaeppi community.** The uptake of every metabolite whose uptake was included during conditional gap filling was blocked separately and the growth rate under this condition was divided by the growth rate for that particular model. A threshold of 0.99 was applied to avoid false-positive responses. Biomass flux ratios were calculated for all metabolic models in the Schlaeppi community. The models were annotated to classes based on the taxonomy published by Bai et al. [38]. The metabolites were grouped based on KEGG BRITE br08001, which was manually refined. Hierarchical clustering was performed using average linkage with Euclidean distance. Cells in dark red indicate low biomass whereas yellow cells indicate a smaller reduction of biomass.

between multiple species. Leaky functions as metabolite diffusion within microbial communities can more likely explain the underlying metabolic interactions and dependencies [6]. The investigation of communities is challenged by the number of participating members and the lack of detailed species-level information.

We investigated the usage of consensus metabolic reconstructions for assessing metabolic interactions between the members of a soil community. This was achieved by devising a novel approach, which fills gaps in the metabolic reconstructions while considering the community composition. More specifically, the obtained solution is dependent on the permeability-based metabolite exchange between the single reconstructions which allows for an investigation of their metabolic dependencies.

First, we demonstrated that the draft reconstructions generated using KBase [13], CarveMe [14], RAVEN 2.0 [17], and AuReMe\Pathway Tools [15] differ substantially from each other, as previously indicated by Mendoza et al. [24]. We hypothesized that the consensus reconstructions generated from the draft reconstructions may resemble manually curated reference models of the same species. We showed that, despite exclusively using automated procedures, consensus reconstructions are highly organism-specific. Some of the reconstructions sharing only the same genus were structurally more similar to the reference than the ones which share the same species. Since in bacteria even strains within a species can differ from each other, the most similar reconstructions of the same genus could indeed also be of the same species or be similar due to metabolic niche adaptation. These results suggests that the draft reconstructions from different approaches complement each other. Further, pairwise Jaccard distances based on different reconstruction features and SVD distance based on the stoichiometric matrices correlated significantly with the phylogenetic distance. These results showed that the differences across the consensus metabolic reconstructions represent the phylogeny of the underlying isolate genomes, providing support for the biological relevance of the reconstructions.

The consensus reconstructions were then used as input to COMMIT to find a minimal set of reactions allowing growth simulations. Here, an iterative procedure applied on individual reconstructions was chosen over solving a gap-filling problem for the whole community because the size of the arising LP would have exceeded the scope for a standard LP solver (i.e. $1.2 \cdot 10^6$ x $9.5 \cdot 10^5$). As a result, COMMIT both shapes and significantly reduces the gap-filling solution. The set of added reactions by COMMIT was much smaller than that obtained by gap filling the reconstructions individually. In addition, we hypothesized that the order in which models are gap-filled corresponds to their roles as 'helpers' or 'beneficiaries'. However, we did not observe a significant correlation between the number of permeable metabolites per model and the order in which reconstructions are considered for gap filling by COMMIT. Therefore, we concluded that the reconstructions that are gap-filled first, in the considered orderings of community members, provide specific compounds to the community that cannot be produced by other members. We did not observe a reduction of biomass flux upon blocking of metabolites unique to the first five models in the optimal gap-filling order. Nevertheless, the permeable metabolites from the first five models contained 11 of the 15 exchanged metabolites, which can be produced and secreted at the lowest cost with respect to the number of added reactions. Hence, the roles of isolates as 'helpers' may be manifested in the gap-filling order by their ability to provide specific metabolites to other members.

We showed that the resulting models can be readily used for subsequent analysis tool as SMETANA [10]. The resulting models for one community are, however, specific to that environment and are, therefore, not functional outside of the community context (i.e. with the minimal medium). We note that the obtained models for an isolate from multiple community compositions can be seen as context-specific models [58] of one isolate blueprint model, which will be more versatile than the models obtained for a single gap-filling run.

To confirm the putative dependencies, we investigated biomass reduction upon unavailability of each compound in the medium. The observed auxotrophies appeared to be not influenced by lineage but by community composition. This is in line with previous observations that auxotrophies can evolve relatively fast, given an environment or community providing the respective compounds, e.g. certain amino acids [59].

Finally, we assessed whether the given communities are, as hypothesized, divided into helpers and beneficiaries according to the Black Queen hypothesis [6]. Even though the differences in secreted metabolites were not large, it could be observed that members of the Mycobacteriaceae are among the isolates that secreted the most metabolites in both communities. Moreover, there were differences between the numbers of imported metabolites between bacterial families, which supports the existence of 'helpers' and 'beneficiaries' within the community.

By the community-dependent generation of functional metabolic models, we could show that prediction of metabolic capabilities as well as insight into the community structure can be obtained by solely using automated tools. In conclusion, this study shows that the usage of automatically-generated metabolic models provides a powerful means to analyse large-scale microbial communities even of uncharacterized species.

## Materials and methods

### Genomic data

The 432 genomes that were used for genome-scale metabolic model reconstruction were downloaded from At-SPHERE (http://www.at-sphere.com) [38]. The phylogeny information as well as the sequences from bacterial culture collections were also obtained from At-SPHERE. Abundances of the single isolates in the environmental samples were computed using 16S rRNA sequences that were kindly provided by Ruben Garrido Oter (Max-Planck-Institute for plant breeding, Cologne). For this purpose, the USEARCH v11 software [60] was used to generate an OTU table, which was in turn normalized using the R implementation of the cumulative sum of squares (CSS) method [61]. As an intermediate, the BIOM format [62] was used, and conversion was performed by the respective Python and R implementations.

### Generation of draft reconstructions

The selected approaches, namely: CarveMe [14], KBase [13], RAVEN 2.0 [17], and AuReMe/Pathway Tools [15,16], use different annotation methods and databases which increases the variability of the automatically generated reconstructions. For the CarveMe and RAVEN 2.0 draft reconstructions, the structural annotation from At-SPHERE was employed. CarveMe uses a manually curated, universal bacterial model in which BiGG reactions are linked to sequences. Upon sequence comparisons to the provided reference, reactions with low sequence similarity score of their associated enzymes are removed from the universal model. This approach, thus, guarantees an initial functional model given the medium that was used during the reconstruction. The medium that was used to create the models was, however, removed in a later step to start the gap-filling procedure with a minimal medium. Moreover, draft reconstructions generated using CarveMe contain a universal biomass reaction, as suggested by Xavier et al. [63].

The RAVEN 2.0 reconstruction pipeline for KEGG-based reconstructions constructs multiple sequence alignments (MSA) for all KEGG Orthology (KO) sequences, which were filtered for phylogenetically close sequences. Hidden Markov Models (HMM) were computed based on the MSAs and subsequently searched for in the genome using HMMer v3.2.1 [64]. In addition, a MetaCyc-based reconstruction was generated based on sequence comparison to a

complete model of the MetaCyc reaction database. Both reconstruction types were created and merged for every isolate using functions of the RAVEN 2.0 toolbox.

To reconstruct draft reconstructions using KBase, the genome sequences were annotated using RAST [65] with default parameters. To this end, we used the nucleotide sequences of the assemblies. The functional annotation from RAST is linked to reactions in the ModelSEED reaction database, allowing for the reconstruction of a draft metabolic model. The draft reconstructions were then downloaded from KBase for further processing.

Further, AuReMe/Pathway Tools was used as a fourth reconstruction method. These reconstructions were generated based on data files generated by Pathway Tools [16]. As a functional annotation is required as an input for the Pathway Tools pipeline, all genomes were annotated using DFAST [66] using default parameters. The resulting draft reconstructions from AuReMe contained reactions based on the MetaCyc database.

## Comparison of draft reconstructions

Metabolic models can be compared structurally using similarity measures that either use the stoichiometric matrix itself of particular features of the reconstruction (e.g. reaction or metabolite identifiers). The definitions of all distance measures that were used are described in S1 Text. First, the distance matrices were combined per isolate by finding a compromise matrix using the STATIS method [40–42]. The STATIS method first compares the considered distance matrices by calculating the $R_v$-coefficients [67]. Then, weights are derived from the scaled first eigenvector of the resulting distance matrix, which are finally used to calculate the compromise matrix as a weighted average. Additionally, the Jaccard distance of gene identifiers was calculated between all four methods and the consensus for each isolate. As a second step, all resulting gene Jaccard distances and compromise matrices were again joined using STATIS [40–42].

## Generation of consensus reconstructions

As a first step, the draft reconstructions from all approaches were converted to have the same number of fields, which in turn share a common format. This is particularly important for reactions, metabolites, genes, and gene-protein-reaction (GPR) rules. Reaction and metabolite identifiers were translated to MNXref identifiers using the MNXref reference files for metabolites and reactions. If a periplasmic compartment was present, all metabolites and reactions were treated as extracellular. Gene identifiers of draft reconstructions generated using KBase and AuReMe/ Pathway Tools were translated by BLAST+ [68] search against the respective structural annotation obtained from At-SPHERE. For unification reasons, also biomass and exchange reactions, if present, were removed.

## Comparison to reference models

The isolates in At-SPHERE [38] were partly taxonomically resolved to species level, among them *Bacillus megaterium* and *Methylobacterium extorquens*, for which manually-curated models had been published. The models for *B. megaterium* WSH-002 [46] and *M. extorquens* AM1 [45] were downloaded and their features were translated to MNXref name space whenever possible. The comparison was hampered by the lack of reaction cross-references to name spaces other than self-defined identifiers. True positives were defined as those identifiers of the consensus reconstruction that were also present in the respective reference. In contrast, false positives were defined as the difference between the consensus and the reference model. Conversely, the difference between the reference model and the consensus comprised the false

negatives. Since the true negatives could not be determined, only sensitivity and precision were calculated as measures to assess the quality of model prediction.

The genomes of the two reference models were downloaded from NCBI reference numbers CP003017 and CP001510.1. MAFFT online service [69] was used to generate MSAs of the 16S rRNA sequences of the isolates and the respective reference, and for NJ/UPGMA phylogeny computation to generate a Newick tree that was converted into a distance matrix using Newick Utilities [70]. Both MAFFT and Newick Utilities were used with the default parameter set.

## Biomass reaction and media

After the merging draft reconstructions from the four different approaches, a universal biomass reaction was added to the reconstructions. The metabolites and their coefficients in the biomass reaction were adopted from the CarveMe reconstructions [14] as they include the universal biomass reaction suggested by [63]. In addition, the initial medium was considered by adding exchange reactions to the reconstructions. The initial medium originates from the intersection of predicted auxotrophic media for the given community obtained from KBase, without organic compounds (except D-glucose as single carbon source).

## Prediction of permeability

To predict metabolite permeability, chemical properties, namely: molecular weight (MW), polar surface area (TPSA), number H-bond donors (HBD), number of H-bond acceptors (HBA), rotatable bonds (RB) and predicted octanol/water partition coefficient (XLogP3), were obtained from PubChem [51] for all metabolites in the MNXref database (S7 and S8 Figs). This was achieved by translating a unique set of 451219 associated InChI keys to PubChem CIDs, whenever possible (116458 out of 451219: 26%), so molecular properties for 110266 compounds could be obtained. Since not all entries had an associated XLogP3 value (only 11894: 11%), the missing values were predicted using k-Nearest-Neighbor (kNN) regression with k = 1 using MW, TPSA, HBD and RB as predictors (S9 Fig) (10-fold cross validation (CV): $R^2_{adj} = 0.87$; $RMSE = 1.26$) (validation set: $R^2_{adj} = 0.9$; $RMSE = 1.11$). The value for k was determined by comparing the performance of multiple kNN models with $1 \leq k \leq 20$ (S9 Fig).

The chemical properties were used to classify metabolites into likely permeable and otherwise by applying Lipinski's rule of five and including RB and TPSA [52,53]. Precisely, the applied rules were: $HBD \leq 5$, $HBA \leq 10$, $MW \leq 500$ Da, $TPSA \leq 140$ Å, $RB \leq 10$, $-0.4 \leq XLogP3 \leq 5.6$.

If no properties were available, the respective metabolite was classified as not likely to be permeable. The above-described classifier was applied to all metabolites in the gap-filling database, which is depicted in more detail below. In total, 8354 out of 33547 (24.9%) metabolites were predicted to be likely permeable. This set of highly-permeable metabolites was enriched with carbohydrates and fatty acids, but also nucleic acids and peptides (p<0.001, ChEBI metabolite ontology [71]). The p-values were corrected for multiple testing using the Benjamini-Hochberg procedure (Table E in S1 Text). Out of 4520 transport reactions, 2766 included at least one highly-permeable metabolite.

## COMMIT formulation

To perform gap filling of the generated reconstructions we implemented he FastGapFilling algorithm [72] in MATLAB and used specific weights for transport reactions (100), metabolic reactions (non-transport, 50), changing a reaction's directionality (25), reactions including

highly-permeable metabolites (subset of transport reactions, 50), reactions with sequence support (25), allowed uptake reactions for metabolites that are secreted by other members (1) and exchange reactions ($10^5$). The penalties applied in this study resulted from the comparison of different weights that were altered until satisfying results were obtained for selected reconstructions, i.e., uptake reactions for secreted metabolites are preferred over the addition of internal reactions and transport reactions for non-permeable metabolites. The upper limit for the biomass flux for each isolate model was constrained to 2.81 h$^{-1}$ as this is the observed growth rate for very fast growing bacteria as Vibrio natriegens [73]. The latter was used to avoid the flux through the biomass reaction being pushed to its default limit of 1000, which would allow a higher number of reactions to be added. Additionally, we added a lower limit to the growth rate, which was set to $10^{-3}$ since it has been reported that soil microbia can have doubling times up to 100 h, which related to a growth rate of $6.9 \cdot 10^{-3}$h$^{-1}$ [74]. The additional constraints on biomass are, however, not propagated to the resulting model, as it is only part of the gap-filling LP.

The chemically-balanced part of the MNXref database was used as a gap filling database, including 25,740 reactions and 18,675 metabolites, after removing reactions according to the criteria described below. To this end, all generic compartments were ignored since a unique translation to either cytosol or extracellular space was not possible. Further, the metabolite ID 'MNXM01' was replaced with 'MNXM1' since both encode H$^+$. Additionally, export reactions i.e. reactions that include metabolites associated to the '@BOUNDARY' compartment were excluded from the gap-filling database. Reactions including stoichiometric coefficients above 20 were also removed.

To determine weights that quantify sequence similarity, HMMs constructed for all KO files were queried against every genome using HMMer [64]. The KO identifiers were matched to MNXref reactions via associated KEGG reactions. An E-value of $10^{-6}$ was used as a cut-off. A lower weight was also assigned to transport reactions that include highly-permeable metabolites as indicated above. To this end, a subset of transport reactions that use highly-permeable metabolites was found in the gap-filling database. Further, one can choose to include sink reactions for cytosolic metabolites in the objective of the gap-filling LP. A sink reaction is an artificial reaction in the network that draws one or more metabolites from the system. An additional matrix for sink reactions R$_{sink}$ is then created for metabolites that are predicted to be highly-permeable or take part in a transport reaction in the model. These sink reactions can also be assigned a specific weight. Since a weighted sum of biomass flux and flux through the added reactions is maximized, only those sink reactions will be added that do not largely decrease the biomass flux. The sink reactions that were found during gap-filling will not be added to the model. Instead, export reactions from the cytosol to extracellular space as well as exchange reactions for the respective extracellular compounds are added to the model.

For the conditional gap filling, 100 random orderings of the considered reconstructions were inspected during COMMIT. The goal is to identify the best ordering with respect to specified criteria. More specifically, the reconstructions are gap-filled following a given ordering. To this end, every iteration starts with a minimal medium, which is augmented by the exportable metabolites of the gap-filled reconstructions. The medium for the first reconstruction in each iteration comprises metabolites that render the particular model auxotrophic. This is used so as to avoid the gap filling of the first reconstruction in the ordering to be unrealistically large in comparison to solutions for subsequent reconstructions. The auxotrophy media were obtained from KBase using the corresponding function. To arrive at a minimal medium, only nutrients were considered that were found in all auxotrophic media of the considered organisms, excluding amino acids and carbon sources, except for glucose. After each single gap-filling run, exportable metabolites are predicted and made available to the subsequent

reconstructions via additional uptake reactions in the gap-filling database. The prediction of exchanged metabolites can be done during the gap filling step of the individual reconstruction or right after. When the first option is used, sink reactions for cytosolic metabolites are integrated directly into the objective of the gap-filling algorithm. Otherwise, a similar LP is solved using the gap-filled model maximizing the flux through sink reactions without reducing the biomass production by more than the given factor. For the second option, the optimal biomass flux $v_{bio}^{opt}$ is determined first, followed by the maximization of flux through the sink reactions while guaranteeing a sub-optimal biomass $v_{bio}^{opt*} = f \cdot v_{bio}^{opt}$. In this study, f = 0.9 has been used (please see S1 Text for more detail). Further, compounds that have an uptake in the model reaction are not reported by this method.

Linear Program for the inclusion of sink reactions after gap filling:

$$\max \sum_{i \in R_{sink}} v_i$$

**s.t.**

$$Sv = 0$$

$$0 \leq v_i \leq 1000$$

$$v_{bio} \geq v_{bio}^{opt*}.$$

Please, refer to S1 Text for the full specification of the COMMIT procedure.

## Supporting information

**S1 Fig. Structural and sequence similarity of consensus reconstructions and isolate sequences to selected reference models. (A)** The sensitivity (left) and precision (right) with respect to metabolite and E.C. number sets were calculated for each of the 432 reconstructions based on the two reference models. **(B)** Sequence similarity of 16S rRNA sequences of isolates to the ones of the two reference species. The red line depicts the median and the box limits represent the 25% and 75% quartiles, respectively. The black dots indicate isolates that were assigned the same genus (9 for *Bacillus megaterium* and 27 for *Methylobacterium extorquens*) according to Bai et al. [38]. Isolates that were predicted to belong to the same species are shown as red dots (2 for *B. Megaterium* and 5 for *M. extorquens*).
(PNG)

**S2 Fig. Similarity of draft metabolic reconstructions to selected reference models.** The sensitivity (left) and precision (right) with respect to metabolite and E.C. number sets were calculated for each of the 432 reconstructions based on the two reference models. These values were scaled by the sequence similarity to the 16S rRNA sequences of the used references. **(A)** KBase [13] **(B)** CarveMe [14] **(C)** RAVEN 2.0 [17], and **(D)** AuReMe/Pathway Tools [15,16]. The black dots indicate isolates that were assigned the same genus (9 for *Bacillus megaterium* and 27 for *Methylobacterium extorquens*) according to Bai et al. [38]. Isolates that were predicted to belong to the same species are shown as red dots (2 for *B. Megaterium* and 5 for *M. extorquens*).
(PNG)

**S3 Fig. Precision and recall of recovered reactions by COMMIT after removing reactions from a two-species community.** COMMIT was run with (cond.) and without (ind.)

consideration of the community composition, i.e. taking permeable metabolites into account. The procedure of removing random reactions (1, 2, 5, and 10 percent) was repeated 50 times for **(A)** the Desulfovibrio vulgaris [56] and **(B)** the Methanococcus maripaludis [55] metabolic models. We only allowed flux-carrying, internal reactions to be removed, which could also be translated to the MNXref namespace (341 for D. vulgaris and 304 for M. maripaludis). (PNG)

**S4 Fig. Comparison of gap filling solutions from COMMIT and individual models for the Bulgarelli community.** Full consensus (consensus), consensus without CarveMe reconstructions (-CarveMe), and KBase draft reconstructions (KBase) were gap-filled either individually or using the COMMIT approach. (A) Sizes of gap-filling solution sets were compared for each reconstruction type using a paired Wilcoxon rank sum test (*** p<0.001). (B) Pairwise comparison of added reactions obtained for each reconstruction and gap-filling type by calculating the Jaccard similarity per isolate. The resulting matrices were merged per group using the STATIS method [40–42]. The obtained values were grouped using K-means clustering (K = 3). The line types indicate the average similarity between the compared groups. (PNG)

**S5 Fig. Putative metabolic interactions between the bacterial families in the Bulgarelli community.** The sets of imported and secreted metabolites that were determined for each member during conditional gap-filling and grouped into corresponding bacterial families. On the left-hand side, metabolite export is shown, which does not require a reduction of growth greater than 10%. The common pool of exchanged metabolites is shown in the center, which were classified using the KEGG BRITE br08001 with manual refinement. On the right-hand side, import reactions are shown, which were introduced during the gap-filling procedure. Line widths represent the number of community members per import/export of a metabolite, scaled by the abundance of the respective family. (PNG)

**S6 Fig. Agreement between predicted metabolic interactions of COMMIT and SMETANA in the Schlaeppi community.** An interaction between two community members was defined as non-empty overlap between the respective sets of imported and exported metabolites. As a result, we obtained a directed graph, in which we scored whether an edge was present with both (green), either one of the methods (light or dark blue) of none of the methods (orange). The directed exchanges returned by SMETANA [10] (implementation from github.com/cdanielmachado/smetana) were transformed to undirected interactions by taking the pairwise overlap of imported and exported metabolites for each pair of reconstructions. (PNG)

**S7 Fig. Correlation of molecular properties for metabolites in the MNXref biochemical database.** The correlation between molecular weight (MW), polar surface area (TPSA), complexity (CPX), heavy atom count (HAC), numbers of H-bond donors (HBD), H-bond acceptors (HBA) and rotatable bonds, and the predicted XlogP3 values are shown as scatter plots in the upper right triangle and the pearson correlation is given in the lower left triangle. (PNG)

**S8 Fig. Distributions of molecular properties obtained from PubChem.** Histograms of the molecular properties obtained for the metabolite in the MNXref database: molecular weight (MW), polar surface area (TPSA), complexity (CPX), heavy atom count (HAC), numbers of H-bond donors (HBD), H-bond acceptors (HBA) and rotatable bonds, and the predicted

XlogP3 values. The density line is shown in red.
(PNG)

**S9 Fig. Prediction quality of different kNN regression models.** The root mean square error (RMSE) and adjusted R-squared value were compared between kNN regressions with different values for **k**. The utilized regression model with k = 1 is highlighted in light grey.
(PNG)

**S1 Text. Supplementary Methods and Tables.** Detailed information on the generation of draft reconstructions, distance measures for model comparison, consensus generation, and the pseudocode of the COMMIT procedure. **Table A in S1 Text. Abundances and numbers of exchanged metabolites of all models in the Schlaeppi community.** The abundances were summed up between the environmental samples investigated in the study after normalization [36]. The numbers of imported and exported metabolites are given as their set difference to the medium. Further, only highly-permeable metabolites were considered, respectively. **Table B in S1 Text. Abundances and numbers of exchanged metabolites of all models in the Bulgarelli community.** The abundances were summed up between the environmental samples investigated in the study after normalization [37]. The numbers of imported and exported metabolites are given as their set difference to the medium. Further, only highly-permeable metabolites were considered. **Table C in S1 Text. Minimal medium used for gap filling.** Medium, which has been used for conditional and individual gap filling of the metabolic reconstructions used in this study. It is composed of nutrients that were predicted as required by all models used in both communities. **Table D in S1 Text. Quality assessment using the MEMOTE test suite** [75]. Unfortunately, we were not able to obtain memote scores for RAVEN 2.0 reconstructions. **Table E in S1 Text. ChEBI metabolite ontology enrichment of the highly-permeable metabolites in the gap-filling database.** An enrichment analysis of ChEBI metabolite ontology terms was used to compare metabolites, which have been predicted to be highly permeable were compared to all metabolites in the gap-filling database. The p-values have been corrected for multiple testing using the Benjamini-Hochberg procedure. The terms are further sorted by their occurrence in the highly-permeable metabolite set.
(PDF)

## Acknowledgments

P.W. and Z.N. would like to thank the Research Focus Group "Evolutionary Systems Biology" of University of Potsdam for support. Z.N. would like to thank the Max Planck Society for support.

## Author Contributions

**Conceptualization:** Philipp Wendering, Zoran Nikoloski.

**Data curation:** Philipp Wendering.

**Formal analysis:** Philipp Wendering, Zoran Nikoloski.

**Investigation:** Philipp Wendering.

**Methodology:** Philipp Wendering, Zoran Nikoloski.

**Project administration:** Zoran Nikoloski.

**Resources:** Zoran Nikoloski.

**Software:** Philipp Wendering.

**Supervision:** Zoran Nikoloski.

**Visualization:** Philipp Wendering.

**Writing – original draft:** Philipp Wendering, Zoran Nikoloski.

**Writing – review & editing:** Philipp Wendering, Zoran Nikoloski.

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
