## [Decision Letter · Decision Letter 0]

16 Oct 2021

Dear Dr. Nikoloski,

Thank you very much for submitting your manuscript "COMMIT: Consideration of metabolite leakage and community composition improves microbial community models" for consideration at PLOS Computational Biology.

As with all papers reviewed by the journal, your manuscript was reviewed by members of the editorial board and by several independent reviewers. In light of the reviews (below this email), we would like to invite the resubmission of a significantly-revised version that takes into account the reviewers' comments.

We cannot make any decision about publication until we have seen the revised manuscript and your response to the reviewers' comments. Your revised manuscript is also likely to be sent to reviewers for further evaluation.

Sincerely,

Vassily Hatzimanikatis

Associate Editor

PLOS Computational Biology

Kiran Patil

Deputy Editor

PLOS Computational Biology

Reviewer's Responses to Questions

**Comments to the Authors:**

Reviewer #1: The authors present a tool that gap-fills individual microbial metabolic reconstructions based on the metabolic reconstructions of organisms found in the same community. The COMMIT algorithm starts with gap-filling a single random metabolic model from the community on minimal media. COMMIT takes into account the permeability of metabolites when deciding on the set of gap-filled reactions. After finding the optimal gap-filling reaction set, it simulates the maximum biomass flux of the model and adds the list of secreted metabolites to the media for gap-filling the next randomly chosen model, and so on, until all models have been gap-filled. COMMIT metabolic reconstructions contain fewer gap-filled reactions than their individually gap-filled counterparts from other sources. COMMIT is a novel gap-filling algorithm that considers both metabolite permeability and other members of the microbial community, which are important factors to consider when modeling microbial metabolism. While I see many uses for this method, I also have a few concerns and questions about the general applicability of the metabolic reconstructions that are produced by the algorithm.

Major points

If you have an organism that is found in multiple different communities, how different is the resulting metabolic reconstruction? With COMMIT, the gap-filling reactions are dependent on the community, so supposedly the end-result would differ. With that in mind, can a COMMIT reconstruction of an organism be used in different applications or only within the context of that particular community?

It is not clear how much (if any) of this work is manual. The authors talk about selecting the optimal set of secreted metabolites based on summed growth rate and number of gap-filled reactions, but it is not clear if this is automatically done by the algorithm or if this step is manual.

Some general terminology needs reworking regarding the use of the phrases metabolic model vs. reconstruction. The authors frequently use “model” where “reconstruction” would be more appropriate. In short, a reconstruction is the set of reactions, genes, and metabolites, whereas a model is a reconstruction with applied constraints and is used for simulations.

The authors associate reconstruction quality by the number of gaps, but I disagree with that assessment. Depending on what you want to learn from the reconstruction or its models, it’s good to have gaps as they tell you where there is a lack of knowledge of the microbe’s metabolism.

L112-114: How come KBase, CarveMe and RAVEN 2.0 were more similar to each other? Do they build on the similar annotations, databases, or algorithms? Looking at the lower triangle in Fig. 1A, how come AuReMe is so completely different from the others? They look about 90% different, what is the reason the gene differences are so big? How come RAVEN is so similar to the consensus but the others seem quite distant?

Figure 1B and 1C: I don’t understand these figures; they need clearer explanations. What is the orange Sum boxplot showing? It is not clear from the figure or caption.

How does a consensus model from just the species-specific reconstructions compare to the genus-specific reconstructions (red dots in Fig. 1D and Fig. S1)? Is a consensus model from all isolates useful?

What is the unscaled sensitivity of the reconstructions in Fig. 1D and S1 and how does the sequence similarity compare among all the genomes? I would expect that the genus-specific genomes are less similar than the species-specific ones, so how does that affect the scaling? Is the sensitivity similar among all and it’s the sequence similarity that is dragging the genus-specific reconstructions down in the plots, or a combination of lower sequence similarities and lower sensitivity?

L336-337: The manuscript states a “strong reduction” here, whereas in Fig. 6 the corresponding field is yellow, which according to the caption indicates “a smaller reduction of biomass”. Also, I would expect that the R. thiooxydans model dependency on sulphur was already present in the draft model, is this not the case? I don’t see how gap-filling the reconstruction (i.e., adding reactions) could introduce a dependency; for that, reactions must be removed from the network or directionality changed from reversible to irreversible (usually only the change from irreversible to reversible is done in gap-filling).

L340-342: I feel that this sentence is overstated in the current version of the manuscript. High-confidence models are compared with experimental data of multiple metabolites, both in terms of uptake and secretion, but this kind of benchmarking has not been done in this study. These models are potentially high confidence, but without some sort of validation for all models for multiple metabolites, I feel that the authors cannot make this statement.

Minor points

L130-133: Model quality assessed with number of gaps. Why is Memote not used for assessing model quality here? It’s used later in the manuscript.

L145-147: Why does RAVEN have larger networks and AuReMe smaller? Are they higher/lower in gap-filled reactions? Was the periplasm compartment accounted for in the comparisons like it was in the consensus models?

L160-161: What does “low translation efficiency” refer to? Reaction/metabolite/gene translation into MXNref? Also, what is the reference point for “low” here? 30%? 50%? 70%?

L161-163: keep in mind that both reference models were built using a draft reconstruction combined from KEGG (used by RAVEN) and Model SEED (uses RAST like KBase), and in the case of B. megaterium, a pre-existing network that I cannot access, and the paper is only available in a foreign language, so I cannot check its origin.

L193: The use of exported and exchanged metabolites is confusing. Perhaps rephrasing “exported” metabolites to “secreted” might make the text clearer?

L198: The new definition of “exchangeable metabolite” is unfortunate here given the traditional phrase for a metabolic reconstruction boundary reaction is “exchange reaction”. Is there any other phrase that could make the distinction between the two clearer?

L237-239: I find Fig. 3 generally difficult to understand. It is not clear how the scaling has been done. Which value is being scaled against which value in the optimal solutions?

L240-241: Here comes one example of my confusion with “exchanged metabolites”. In Fig. 3, is the number of exchanged metabolites referring to the number of metabolites secreted by each gap-filled model, or is it the total number of metabolites added to the minimal medium?

L252-253: “suggesting a dependence between these two criteria”. I am not sure I understand this correctly; the results suggest a dependence of the sum of biomass reaction fluxes and the number of exchanged metabolites added to the medium? I would have expected that to be the case, unless maybe if all models were already growing at their maximum capacity.

L260-262: What were the scores of the two reference models? Scores are only (partially) listed for B. metaterium in the text. For an audience not familiar with memote, what are good scores?

L277: Fig. 4B, the line types indicate the average similarity based on what? Reactions?

L525-527: There’s a maximum biomass flux, is there a minimum? Maybe the other end of the range; 2.8 1/h being very fast-growing bacteria, perhaps the other end could be constrained to very slow growing bacteria?

Reviewer #2: In this work, Wendering and Nikoloski develop a community gap-filling algorithm that aims to identify a minimal solution set, while exploring 100 random gap-filling orderings, according to the following criteria: number of added reactions, dependence of the first member on exported metabolites of subsequent models, number of exchanged metabolites, and sum of biomass fluxes of all community members. The authors observed that the solution set obtained by COMMIT was significantly smaller than that obtained by gap-filling each model individually. This is a contribution that adds to a large body of literature that addresses similar challenges for individual models and microbial community ones. There is a number of key issues that remain unclear.

Major comments:

1) It is not surprising that the number of added reactions is lower as it is mathematically provable that allowing for cross-feeding between models will lead to reduction of the need of added reactions for connectivity. What is needed is to quantify whether the reduction reaches a particular threshold that would not have happened by chance. Possibly an F-test would be a solution.

2) The optimization formulation minimizes sum of molar fluxes. This de-emphasizes flows involving metabolites with large molecular weights possibly causing systematic discrepancies.

3) Gap filling algorithms including COMMIT are not unsupervised algorithms for fixing gaps. Each gap resolution strategy in the model needs to be scrutinized for biochemical evidence. In our experience, gap-filling without any follow up leads to bloated models that propagate misannotated reactions in follow up investigations. The authors need to specify specific instances where the cross-feeding and/or strategy suggested by COMMIT was experimentally verified. Also, they need to caution for cases where COMMIT tends to make systematically erroneous suggestions. Absent of this, this effort is simply a computational exercise devoid of any potential for new insight.

4) Gap filling almost always leads to the formation of thermodynamically infeasible cycles especially for community models. It is unclear whether there is a safeguard against this. Are there any unbounded fluxes?

5) The authors need to make the case why we need yet another algorithm for gap filling. What is new in the approach that is beyond a simple modification of existing approaches.

6) L76-78: potentially misleading, as SteadyCom solves a set of linear problems iteratively, where the number of iterations is independent of the number of organisms in the community

7) L423: since CarveMe is stated to generate a GSM model using a supplied medium, not clear how using a static medium is justified in this work as nutrients available to an organism will be a function of metabolites excreted by others in the community and the plant root exudates as well

8) L439-443: why were different genome annotation tools (such as RAST and DFAST) used for different GSM reconstruction tools? How do the results differ if a single tool is used?

9) The authors state that the models generated using automated tools are of high confidence but provide no comparison between simulated growth and experimental data. Have those models been validated? Can the models display growth individually? Or the only growth simulations performed were as a community? Did the predicted results quantitatively match experimental observations?

10) The biomass equation acts as a sink in metabolic models, and thus a primary determinant of network fluxes and capabilities. Since GSM models of two community members is available, how do the results change if the biomass reactions from these models are used instead of the universal biomass reaction from CarveMe reconstructions?

11) The motivation behind determining highly-permeable metabolites and treating them differently during gap-filling is clear, but how would the authors account for possible confoundments that differentiate a permeable metabolite from being available for consumption by other members in a community, such as intracellular concentrations, and role as a metabolic substrate?

12) L379-381: If the ordering of the gap-filling is what assigns a community member to be a ‘provider’ (by providing specific compounds to the community that cannot be produced by other members), can the authors comment on the inferences made about that organism’s role in the microbial community being biased by the gap-filling process?

Minor comments:

1) L54: Suggest replacing ‘analyse’ for ‘analyses’

2) L525: Suggest explaining in further detail what ‘satisfying results’ entail

3) L256: Suggest specifying whether the biomass flux upper bound pertains to the entire community or to each of the individual consensus models

4) L567: It would benefit the readers if the definition of sink reactions was provided somewhere in the text.

Reviewer #3: In this paper, the authors present a new gapfilling tool COMMIT for automated large-scale modeling of microbial communities. This approach allows for gapfilling of consensus models at the community level. The authors then apply the approach to analyze microbial communities associated with Arabidopsis. They construct models for over 400 arabidopsis rhizosphere isolates using multiple model reconstruction approaches. They merge these models into consensus models. The commit approach then gapfills individual models one by one, expanding the gapfilling media based on the capabilities of each previous model gapfilling. This is an interesting and valuable approach as it will scale better than the community-level gapfilling employed by tools like KBase. COMMIT could thus support generation of much larger and more complex community models.

Overall, there is clear merit in the proposed interative gapfilling approach, as it does offer a means of gapfilling larger communities. The consensus approach to model building is more questionable. There is some question as to whether a larger model is necessarily better, and incompatibility and conflicts between different modeling approaches may introduce more problems than it solves.

All this said, many questions arise from the studies, methods, and data presented in the paper:

1.) Note, KBase already has tools for constructing and gapfilling community models, which can be constrained to specific abundances. This could perhaps be mentioned in the introduction. Again, as mentioned above, this approach gapfills the entire community at the same time and thus won’t scale to large numbers of species.

2.) Merging and reconciling models, particularly models constructed from different biochemistry databases like KEGG, ModelSEED, BIGG and MetaCyc can be extremely challenging because of differences in the representation of biochemistry among these sources. Many metabolites don’t have structures and thus aren’t matched well in resources like MNXref. In other cases, reactions differ by the level of lumping performed, the number of monomers in a polymeric compound, or compartmentalization of metabolites in the reaction. Did the authors encounter these challenges? How was this overcome?

3.) Note, in building KBase models, the authors could have retained the AuReMe gene calls and identifiers and just had RAST reannotate the existing genes. KBase permits upload of existing gene calls in GFF and genbank format. This would probably be a preferable workflow for what the authors wanted to do.

4.) Was a single universal biomass used? Why not vary the biomass based on species (e.g. gram positive, gram negative, archaea)? Would the method be sensitive to this? Can the decision to use a single universal biomass be justified?

5.) Did charge (at pH 7) come into play on the permeability prediction? Highly charged compounds would generally not be permeable.

6.) With consensus modeling, there’s a major concern of adding many false positive reactions. Do the authors have some way to testing for this (e.g. comparing with Biolog data)? The comparison to published models performed is good, but data performance would be helpful as well.

7.) The reaction counts from Raven are generally surprisingly high (significantly larger than many published models). This doesn’t necessarily imply improved quality (but it might be). Can the authors perhaps explain what kinds of reactions Raven is annotating that other methods are missing? Are the reactions active? What kind of metabolism is represented?

8.) How much did the annotations of the models agree? Did they generally assign the same genes to the same reactions? Merging conflicting annotations in consensus models could cause problems. How did the authors handle this case?

9.) How did the authors relate 16s data to their isolate genomes for which they built models? By species taxonomy matches? By blasting 16s sequences against isolate genomes?

10.) Were any metrics used to determine if 100 iterations of the COMMIT algorithm was sufficient? Were solutions significantly different based on the order in which the gapfilling was performed?

**Have the authors made all data and (if applicable) computational code underlying the findings in their manuscript fully available?**

Reviewer #1: Yes

Reviewer #2: Yes

Reviewer #3: Yes

PLOS authors have the option to publish the peer review history of their article (what does this mean?). If published, this will include your full peer review and any attached files.

Reviewer #1: No

Reviewer #2: No

Reviewer #3: No
---

## [Decision Letter · Decision Letter 1]

9 Feb 2022

Dear Dr. Nikoloski,

We are pleased to inform you that your manuscript 'COMMIT: Consideration of metabolite leakage and community composition improves microbial community reconstructions' has been provisionally accepted for publication in PLOS Computational Biology.

Best regards,

Vassily Hatzimanikatis

Associate Editor

PLOS Computational Biology

Kiran Patil

Deputy Editor

PLOS Computational Biology

Reviewer's Responses to Questions

**Comments to the Authors:**

Reviewer #1: I would like to thank the authors for answering my questions and concerns in detail and incorporating changes where applicable. Microbial community metabolic modeling is challenging and I think the COMMIT tool will be good resource for community-level modeling efforts. I have no further comments on the manuscript.

Reviewer #2: The authors addressed most of our comments. They may want to caution the reader in regard to the possibility/likelihood of the formation of thermodynamically infeasible cycles and ways to remedy this.

Reviewer #3: The authors have generally adapted the manuscript well to my comments. However, the issue of using a single universal biomass for all species seems like an important one. I would argue that it is the distinctiveness of the biomass, specifically the organism-specific cofactor requirements and thus cofactor availability, that account of alot of the reason for various species to exist in the community (some species can simply do chemistry that others cannot). This is somewhat neglected when using a universal biomass. None of those cofactor biosynthesis pathways or transports will be activated. I understand that correcting for this falls outside the scope of the current study, but some discussion in the revised manuscript of this limitation and the need to improve methods to compensate for it seems important to include. Otherwise, it's likely readers won't even realize a universal biomass is being used as this detail is somewhat buried in the methods.

**Have the authors made all data and (if applicable) computational code underlying the findings in their manuscript fully available?**

Reviewer #1: Yes

Reviewer #2: Yes

Reviewer #3: Yes

PLOS authors have the option to publish the peer review history of their article (what does this mean?). If published, this will include your full peer review and any attached files.

Reviewer #1: No

Reviewer #2: No

Reviewer #3: No

---

## [Editor Report · Acceptance letter]

9 Mar 2022

PCOMPBIOL-D-21-01599R1 

COMMIT: Consideration of metabolite leakage and community composition improves microbial community reconstructions

Dear Dr Nikoloski,

I am pleased to inform you that your manuscript has been formally accepted for publication in PLOS Computational Biology. Your manuscript is now with our production department and you will be notified of the publication date in due course.

With kind regards,

Katalin Szabo
